# CREBBP/EP300 acetyltransferase inhibition disrupts FOXA1-bound enhancers to inhibit the proliferation of ER+ breast cancer cells

Archana Bommi-Reddy☯, Sungmi Park-Chouinard☯, David N. Mayhew¤a☯, Esteban Terzo¤b, Aparna Hingway¤c, Michael J. Steinbaugh🄳¤d, Jonathan E. Wilson¤e, Robert J. Sims, III¤e, Andrew R. Conery🄳¤f *

Constellation Pharmaceuticals, a Morphosys Company, Cambridge, Massachusetts, United States of America

☯ These authors contributed equally to this work.
¤a Current address: Foghorn Therapeutics, Cambridge, Massachusetts, United States of America
¤b Current address: Eloxx Pharmaceuticals, Watertown, Massachusetts, United States of America
¤c Current address: Novartis Institutes for Biomedical Research, Cambridge, Massachusetts, United States of America
¤d Current address: 28–7 Therapeutics, Cambridge, Massachusetts, United States of America
¤e Current address: Flare Therapeutics, Cambridge, Massachusetts, United States of America
¤f Current address: Triana Biomedicines, Waltham, Massachusetts, United States of America
* arconery@gmail.com

**Data Availability Statement:** Datasets generated during this study are deposited at the Gene Expression Omnibus (GEO): GSE190163.

## Abstract

Therapeutic targeting of the estrogen receptor (ER) is a clinically validated approach for estrogen receptor positive breast cancer (ER+ BC), but sustained response is limited by acquired resistance. Targeting the transcriptional coactivators required for estrogen receptor activity represents an alternative approach that is not subject to the same limitations as targeting estrogen receptor itself. In this report we demonstrate that the acetyltransferase activity of coactivator paralogs CREBBP/EP300 represents a promising therapeutic target in ER+ BC. Using the potent and selective inhibitor CPI-1612, we show that CREBBP/EP300 acetyltransferase inhibition potently suppresses in vitro and in vivo growth of breast cancer cell line models and acts in a manner orthogonal to directly targeting ER. CREBBP/EP300 acetyltransferase inhibition suppresses ER-dependent transcription by targeting lineage-specific enhancers defined by the pioneer transcription factor FOXA1. These results validate CREBBP/EP300 acetyltransferase activity as a viable target for clinical development in ER+ breast cancer.

## Introduction

Estrogen Receptor positive breast cancer (ER+ BC) constitutes approximately 60–80% of breast cancer cases and ER signaling is acknowledged as the oncogenic driver of the disease [1]. As such, anti-estrogen therapy is the mainstay of treatment in ER+ BC [2]. Although there is clear evidence that current therapies have prolonged patient survival, a majority of patients

**Funding:** The author(s) received no specific funding for this work.

**Competing interests:** Authors are current or former employees and stockholders of Constellation Pharmaceuticals, a Morphosys company, which provided funding for this research. Patents have been filed around the chemical series that includes CPI-1612. This work does not relate to any marketed products or products in development. These disclosures do not alter our adherence to PLOS ONE policies on sharing data and materials.

with metastatic disease acquire resistance and relapse [3, 4]. Therefore, new therapies are continually required to improve clinical outcomes for these patients.

Cyclic AMP response element-binding binding protein (CREBBP) and its closely related homolog E1A binding protein of 300 kDa (EP300) are ubiquitously expressed multidomain transcriptional coactivators whose histone acetyltransferase (HAT) domains are highly conserved throughout evolution [5]. CREBBP/EP300 catalyze lysine acetylation on a broad range of substrates, particularly K18 and K27 on histone H3 and several non-histone proteins, to regulate signaling pathways involved in cell growth, development, and tumorigenesis [6, 7]. Recently, it has been demonstrated that CREBBP/EP300 act to regulate lineage-specific transcriptional programs (as opposed to global transcriptional programs), which are largely driven by distal enhancers and tissue-specific transcription factors [8].

CREBBP/EP300 are core components of the ER transcriptional complex [9], and acetyltransferase activity of CREBBP/EP300 is critical for ER signaling [10]. Within the complex, CREBBP/EP300 act through direct acetylation of ER to enhance its DNA binding and transactivation function [11]. Beyond this direct activity on ER, one might also hypothesize that ER-bound CREBBP/EP300 act on histones to create regions of hyperacetylation and open chromatin to facilitate the recruitment of transcriptional machinery. ER signaling is known to be reprogrammed through enhancer remodeling during breast tumorigenesis [12]. Further, the majority of ER binding sites (estrogen response elements, or ERE) have been mapped to distal enhancers, consistent with the activity of CREBBP/EP300 [13, 14]. The magnitude of changes in chromatin accessibility and the relationship to the histone acetyltransferase activity of CREBBP/EP300 have not been examined in the context of ER-driven transcription.

We propose that inhibiting the HAT domain of CREBBP/EP300 will be an effective strategy to orthogonally target the clinically validated ER transcriptional network. By not targeting ER directly, therapeutic inhibition of CREBBP/EP300 has the advantage of being active in the context of resistance to anti-estrogen therapies. For many years the only options for targeting CREBBP/EP300 acetyltransferase activity were natural products or nonspecific inhibitors, but recently multiple highly selective, potent, and orally bioavailable CREBBP/EP300 HAT inhibitors have been described, including A-485 [15] and CPI-1612 [16]. Here we use CPI-1612 to demonstrate that inhibition of CREBBP/EP300 HAT activity represents a promising strategy to suppress ER-mediated proliferation and tumor growth. Further, we use this potent and selective inhibitor to provide insight into the role of CREBBP/EP300 in defining the transcriptional programs and chromatin landscape of ER+ breast cancer.

## Results

### CREBBP/EP300 HAT inhibition abrogates ER-driven proliferation *in vitro* and *in vivo*

Given the functional link between CREBBP/EP300 and ER, we investigated whether inhibition of CREBBP/EP300 acetyltransferase activity would impact the viability of ER+ breast cancer cell lines. Pooled, barcoded breast cancer cell lines were treated with a dose titration of the previously described potent and selective inhibitor of CREBBP/EP300 acetyltransferase activity, CPI-1612 [16]. Growth inhibition was measured using barcode depletion after 5-day treatment with compound as described [17]. Consistent with previous results [15], CREBBP/EP300 HAT inhibition shows activity in a subset of triple negative breast cancer (TNBC) cell lines, but we also noticed potent growth inhibition in ER+ cell lines (panel A in S1 Fig). Given the known functional link between CREBBP/EP300 and ER signaling, we further explored the impact of CPI-1612 in the context of ER+ breast cancer.

We first confirmed that CPI-1612 is highly active in ER positive breast cancer cell lines in standard growth inhibition assays, with $GI_{50}$ values below 100 nM (Fig 1A). Importantly, these $GI_{50}$ values are in the same range as published $EC_{50}$ values for reduced H3K18 acetylation by CPI-1612, arguing for on-target effects of CREBBP/EP300 inhibition [16]. To exclude impacts of CPI-1612 on non-hormone driven growth factor networks in full serum, we made use of growth factor/hormone depleted charcoal-stripped media and exogenously added estradiol (CSS+E2). We noted that $GI_{50}$ values for CPI-1612 were similar for ER positive breast cancer cell lines in both culture conditions (panel B in S1 Fig), demonstrating that CREBBP/EP300 inhibition can directly impact hormone-driven proliferation.

To investigate the anti-tumor effects of CREBBP/EP300 HAT inhibition in ER+ breast cancer in vivo, we used an MCF7 xenograft model. CPI-1612 has favorable ADME properties and achieves sufficient exposure to induce pharmacodynamic and anti-tumor responses in vivo [16]. Oral dosing with CPI-1612 twice daily in established MCF7 xenografts resulted in dose-dependent inhibition of tumor growth (Fig 1B), and all doses were well tolerated (panels C and E in S1 Fig). Notably, CPI-1612 treatment led to dose-dependent reduction in H3K27 acetylation in peripheral blood and in H3K18 acetylation in tumor cells, demonstrating target engagement at efficacious doses (Fig 1C; panel D in S1 Fig).

Standard of care (SOC) treatment for ER positive breast cancers consists of anti-hormone therapy such as the selective ER degrader (SERD) Fulvestrant. To determine whether CREBBP/EP300 acetyltransferase inhibition could enhance the response to SOC therapy, we treated xenografted MCF7 cells with Fulvestrant and CPI-1612 alone or in combination. To maximize the potential for combinatorial effects, a suboptimal dose of CPI-1612 was used in combination with a clinically relevant dose of Fulvestrant [18]. As shown in Fig 1D, treatment with low dose CPI-1612 enhanced the anti-tumor efficacy of Fulvestrant in a well-tolerated dosing regimen that was associated with reduction in H3K27 acetylation (Fig 1E; panel F in S1 Fig). Enhanced anti-tumor efficacy was also associated with a more pronounced reduction in the mRNA level of the known ER target *MYC* in tumor tissue, suggesting enhanced engagement of the ER transcriptional network (Fig 1F). As shown in panels G and H in S1 Fig, enhanced efficacy and pharmacodynamics were not the result of increased exposure of either CPI-1612 or Fulvestrant. Taken together, these data demonstrate that selective inhibition of CREBBP/EP300 acetyltransferase activity blocks the proliferation of ER positive breast cancer cells in vitro and in vivo, and the enhanced efficacy in combination with Fulvestrant suggests that acetyltransferase inhibition has the potential to potentiate the effects of direct ER targeting, perhaps through increased engagement of ER transcriptional programs.

## CREBBP/EP300 HAT inhibition inhibits ER-dependent transcriptional programs

To explore the transcriptional events underlying the phenotypic response to CREBBP/EP300 acetyltransferase inhibition, we carried out RNA sequencing analysis of three breast cancer cell lines (MCF7, T47D, and ZR751) treated with CPI-1612 alone or in combination with Fulvestrant. To minimize potential secondary effects of long-term treatment, we treated cells for 6 hours prior to sample preparation. As shown in Fig 2A; panel A in S2 Fig, CPI-1612 treatment has broad, dose-dependent effects on gene expression, with the majority of differentially expressed genes showing downregulation, while Fulvestrant has a modest transcriptional impact. Gene set enrichment analysis (GSEA) with Hallmark gene sets revealed that the top two downregulated gene signatures for both CPI-1612 and Fulvestrant treatment were HALLMARK_ESTROGEN_RESPONSE_EARLY and HALLMARK_ESTROGEN _RESPONSE_-LATE, but CPI-1612 has a broader impact, consistent with the higher number of genes

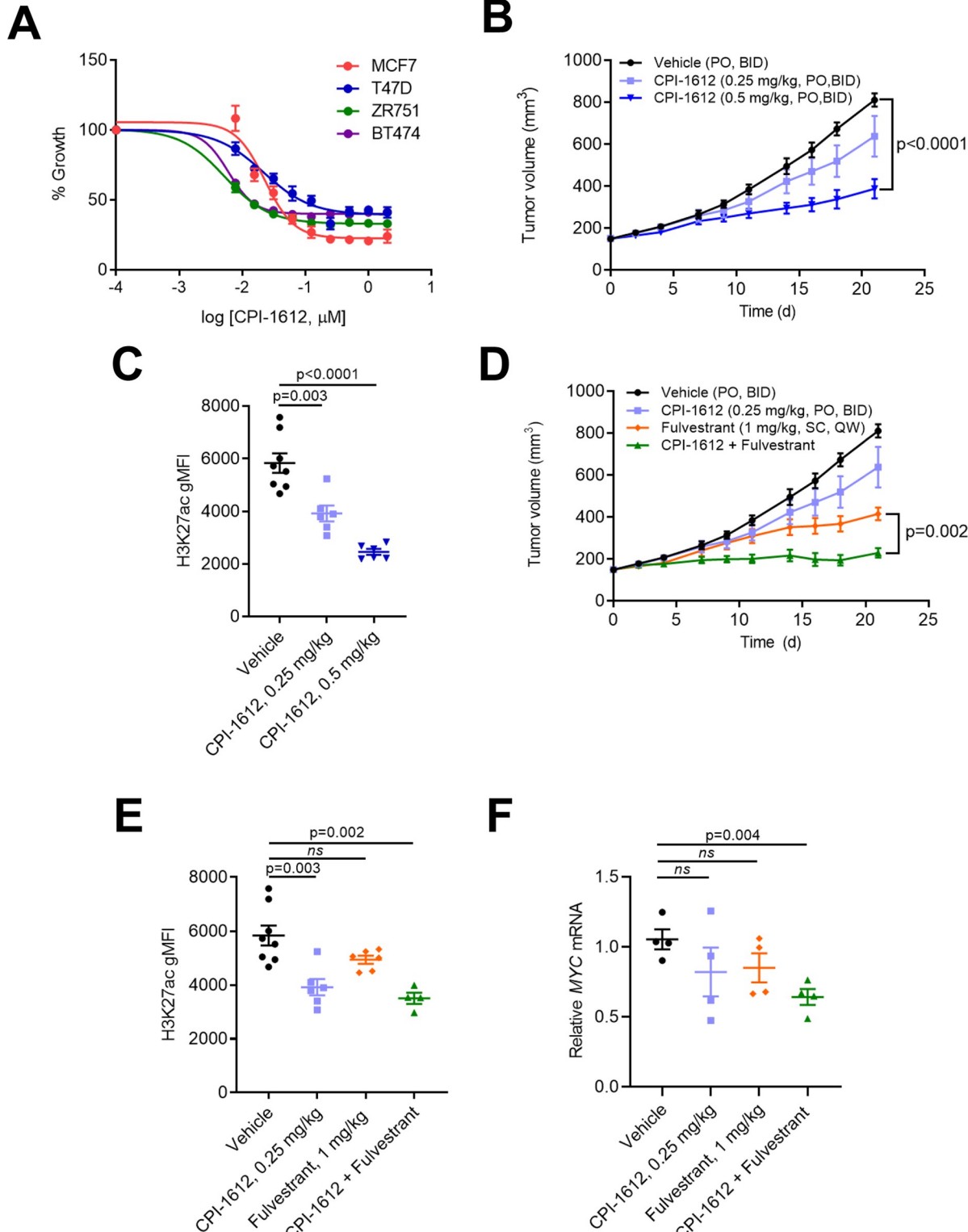

**Fig 1. CPI-1612 inhibits viability of ER+ breast cancer cell lines and ER signaling both in vitro and in vivo.** (A) In vitro activity of CPI-1612. ER+ breast cancer cell lines were treated with increasing doses of CPI-1612 and cell viability was measured using Cell Titer Glo after 4 days of treatment. Error bars represent standard deviation (n = 2). (B) In vivo activity of single agent CPI-1612. Female Balb/c nude mice were implanted subcutaneously with MCF7 cells (n = 8 for vehicle, n = 6 for others) and treated with the indicated doses of CPI-1612 (PO, BID) or an equal volume of vehicle (PO, BID). Tumor volumes were measured by caliper until study termination at 21 days. Data points

represent mean and SEM at each timepoint. P-values were calculated using an unpaired student's t-test relative to the vehicle arm; the p-value at study endpoint is shown (no data point in the 0.25 mg/kg arm reached statistical significance). (C) Pharmacodynamic readout of CPI-1612 activity. PBMCs were isolated from blood at study termination, fixed, and stained for FACS analysis. The level of H3K27ac was quantified using gMFI (geometric mean fluorescence intensity). Data are represented as mean ± SEM, and p-values were calculated using an unpaired student's t-test. (D) Efficacy of CPI-1612 in combination with Fulvestrant. Mice were xenografted with MCF7 cells as in (B) and treated with CPI-1612 (0.25 mg/kg, PO, BID), Fulvestrant (1 mg/kg, SC, QW), CPI-1612 + Fulvestrant, or vehicle (n = 8 for vehicle, n = 6 for others). Data points represent the mean and SEM of surviving animals, and p-values were calculated at each timepoint using an unpaired student's t-test. P-value for the difference between Fulvestrant and CPI-1612 + Fulvestrant at study endpoint is shown. (E) PD in PBMCs for study described in (D), as in (D). (F) Tumor PD as measured by gene expression changes. Total mRNA was isolated from tumors collected at study endpoint and used for q-RTPCR analysis. *MYC* expression normalized to *ACTB* was calculated relative to vehicle mean and is expressed as mean ± SEM for each arm. P-values were calculated by unpaired student's t-test relative to vehicle.

modulated (Fig 2B; panel B in S2 Fig). Gene expression changes showed good overlap among the three ER+ cell lines at the gene level (panel A in S3 Fig), but at the gene signature level the responses were nearly identical (panel B in S2 Fig).

Comparison of the enrichment plots for single agent CPI-1612 (5 nM) or Fulvestrant with combination treatment shows enhanced repression upon combination treatment (panel C in S2 Fig). Investigation of the genes driving gene set repression demonstrates that while both CPI-1612 and Fulvestrant target the ER transcriptional network, they do so by targeting non-identical sets of genes (Fig 2C). For example, while known ER target genes such as *MYC* and *GREB1* are regulated by Fulvestrant or by both treatments, CPI-1612 uniquely regulates a set of genes including *ELF3* and *HES1* (Fig 2C and 2D; panel D in S2 Fig). To compare the genomic and epigenomic features underlying these distinct gene expression changes, we examined these loci for proximal sites of occupancy by ESR1 and EP300, as well as proximal ERE sequences [13, 19, 20]. We noted that while *MYC* and *GREB1* have proximal ESR1 and EP300 binding, *ELF3* and *HES1* have proximal EP300 binding but lack proximal ESR1 binding, consistent with the observed differential gene expression (panel B of S3 Fig). More broadly, we found that EP300 occupancy does not differentiate those genes regulated by Fulvestrant or CPI-1612 (panel C, left, of S3 Fig), but that genes regulated by Fulvestrant are more likely to be bound by ESR1 than those regulated by CPI-1612 (panel C, right, of S3 Fig). Consistent with these ChIP-seq data, genes that are acutely downregulated by Fulvestrant are largely defined by the presence of a proximal ERE, while genes acutely downregulated by CREBBP/EP300 HAT inhibition generally lack a proximal ERE and thus are likely defined by alternative features.

## CREBBP/EP300 inhibition targets a subset of enhancers that are linked to differentially expressed genes

To better understand how CREBBP/EP300 HAT inhibition leads to transcriptional changes in ER+ breast cancer cells, we performed ATAC-seq and H3K27 acetyl ChIP-seq upon treatment of MCF7 cells with CPI-1612. As expected from published substrate profiling of CREBBP/EP300 [7, 21] and from the data shown in Fig 1C, CPI-1612 led to a profound loss of H3K27ac, with more than one-third of all peaks showing at least a two-fold decrease in CPI-1612 treated cells relative to the DMSO control (Fig 3A and 3B). As a control, there were minimal effects on H3K9ac, arguing for the selectivity of CPI-1612. Intriguingly, ATAC-seq results showed that effects on chromatin accessibility were much more modest than changes in H3K27ac, with only about 2% of all open peaks showing the same two-fold change in magnitude, with most differential peaks showing a reduction in accessibility (Fig 3A and 3B). As expected, total ATAC-seq peaks show a high overlap with total H3K27ac peaks (panel A in S4 Fig). However, while the majority of differential ATAC-seq peaks also show differential H3K27ac, the converse is not true, as <5% of differential H3K27ac peaks show significant

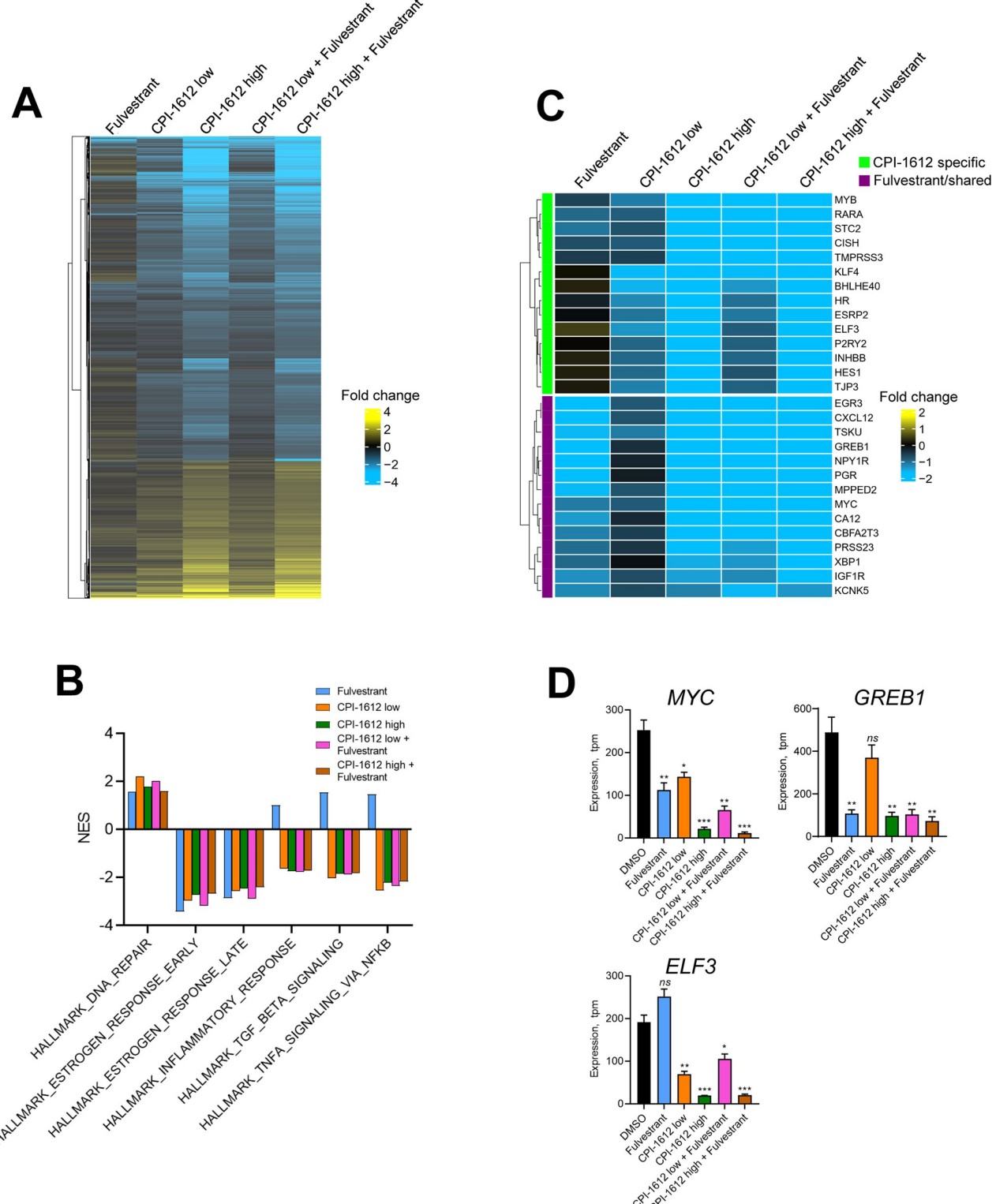

**Fig 2. CPI-1612 inhibits the ER transcriptional program.** (A) Bulk RNA-seq analysis of MCF7 cells. MCF7 cells were treated as indicated for 6 hours (n = 3 per treatment) followed by isolation of mRNA for RNA-sequencing analysis. Differential expression is indicated as $\log_2$ (fold-change) in normalized counts relative to the DMSO control. Genes shown were modulated at least 1.5-fold in at least one condition. Concentrations used were CPI-1612 low: 5 nM; CPI-1612 high: 50 nM; Fulvestrant: 100 nM. (B) CPI-1612 has a distinct transcriptional effect from Fulvestrant. Gene Set Enrichment Analysis (GSEA) was carried out using the MSigDB Hallmark genesets with the data described in (A). Normalized enrichment scores

(NES) for selected genesets are shown. (C) CPI-1612 regulates the ER transcriptional network by impacting different genes than Fulvestrant. Subset of data in (A) showing selected genes in the HALLMARK_ESTROGEN_RESPONSE_EARLY geneset. Genes that are regulated by CPI-1612, Fulvestrant, or both are highlighted. (D) Example of differentially regulated genes. Expression values from RNA-seq are quantified as transcripts per million (tpm) and are plotted from the experiment described in (A). Values represent the mean and SEM for 3 replicates. P-values were calculated by unpaired student's t-test (*: p<0.05; **:p<0.01;***:p<0.001; ns: not significant). P-values can be found in S3 Data.

changes in chromatin accessibility (panels C and D in S4 Fig). Thus, while CREBBP/EP300 HAT inhibition globally reduces H3K27 acetylation in ER+ breast cancer cells, changes in chromatin accessibility are much more circumscribed.

To determine if differentially accessible and acetylated sites were enriched in specific genomic regions, we annotated peaks with nearby genomic features using HOMER [22]. Both H3K27ac and ATAC-seq differential peaks were more likely to be found in intergenic regions and less likely to be found in promoters (Fig 3C). We next checked whether the genes linked to these enhancers were differentially expressed after CPI-1612 treatment by mapping peaks to genes using an approach described previously [23]. Given the relatively smaller number of peaks, the ATAC-seq mapped gene list was smaller than the H3K27ac ChIP-seq mapped list (Fig 3D), yet both gene sets identify several key genes which were down-regulated after CPI-1612 treatment, including *MYC* and *ESR1*, both of which we confirmed to be reduced at the protein level as well as the mRNA level (S5 Fig). GSEA of the genes mapped to differential ATAC-seq or H3K27ac peaks further showed enrichment of the ER transcriptional network (Fig 3E). The transcription factor *ELF3*, a member of the core transcriptional regulatory circuitry in MCF7 cells [24, 25] was one of the most robustly down-regulated genes and is illustrative of the broad reduction of H3K27ac and more focal reduction in chromatin accessibility at distal enhancer sites (Fig 3F).

## CREBBP/EP300 inhibition targets FOXA1 cell type-specific binding sites that control ER signaling and luminal-specific gene sets in breast cancer cells

The large-scale decrease in H3K27ac signal was in line with expectations for inhibition of CREBBP/EP300, but it was less intuitive as to why a much smaller subset of enhancers would change chromatin accessibility status when profiled by ATAC-seq. To test whether these regions contained the motifs for any specific transcription factors, we employed HOMER's findMotifs function [22] to test whether the underlying DNA sequences of the subset of ATAC-seq peaks which close upon CPI-1612 treatment were enriched for any known transcription factor motifs. Strikingly, the FOXA1 transcription factor motif was the most statistically significant motif in this region, with other FOX family motifs also showing significant enrichment, likely due to sequence similarity (Fig 4A). In contrast, differential H3K27ac peaks did not show enrichment of FOXA1 motifs, and only modest enrichment of any motifs (panel A in S6 Fig). To further confirm these observations, we independently performed single-cell ATAC-seq in MCF7 cells treated with either CPI-1612 or DMSO. The single-cell analysis revealed some heterogeneity in the epigenetic composition of MCF7 cells (panel A in S7 Fig); however, in concordance with the bulk ATAC-seq data, most regions of open chromatin did not change after treatment (panel B in S7 Fig). We used chromVAR [26] to identify the transcription factor motifs that had the largest change within variable peaks after treatment. Commensurate with the bulk ATAC-seq data, the motif for FOXA1 showed the largest decrease in predicted binding activity after treatment (panels C and D in S7 Fig).

FOXA1 is a pioneer transcription factor that can open chromatin and facilitate the binding of other transcription factors, including ER, in breast cancer cell lines [27, 28]. FOXA1 binding

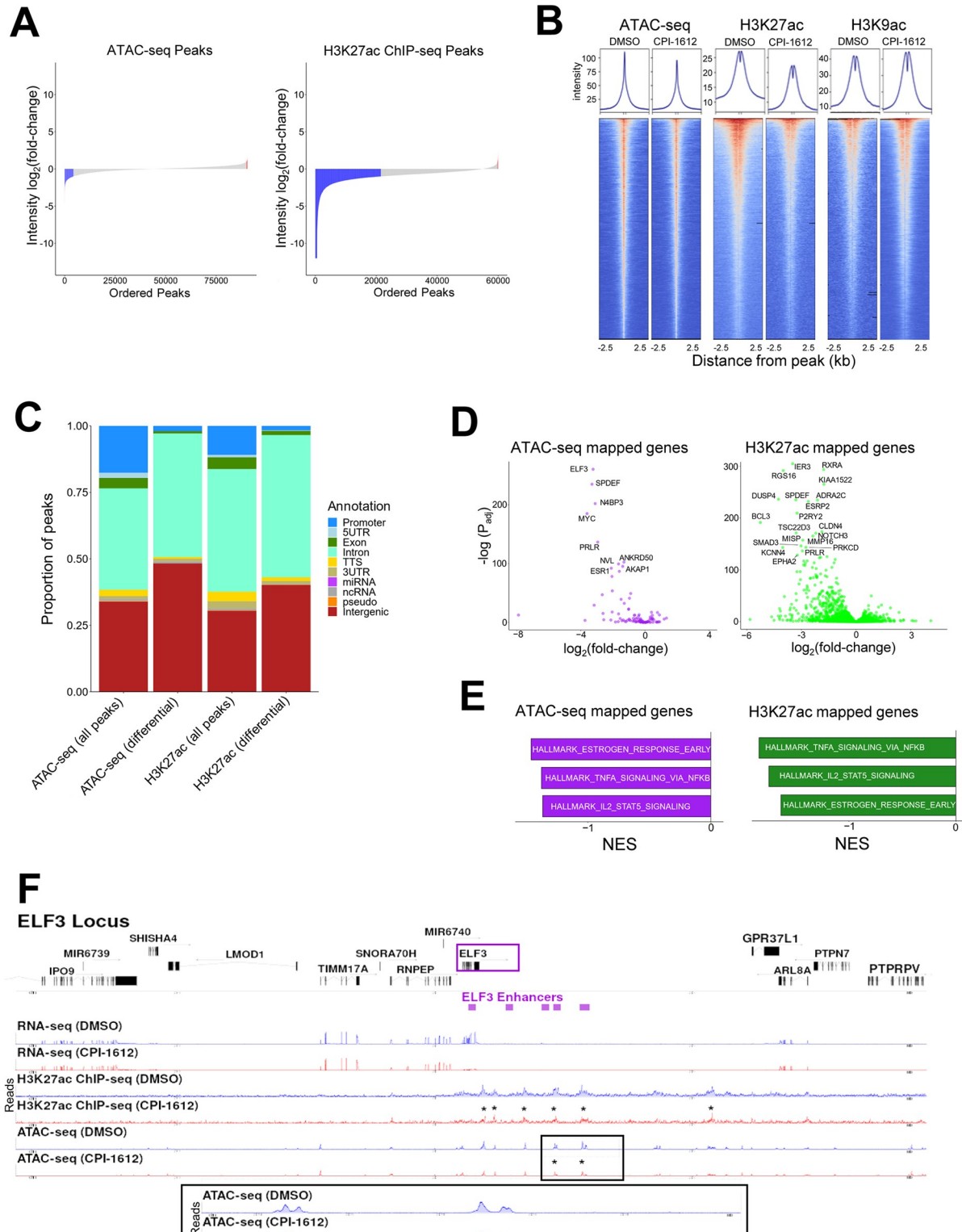

**Fig 3. CPI-1612 represses a subset of enhancers which are linked to differentially expressed genes.** (A) CPI-1612 treatment impacts chromatin accessibility and histone acetylation. MCF7 cells were treated for 6 hours with DMSO or CPI-1612 (50 nM), and samples were prepared for ATAC-seq or ChIP-seq with H3K27ac or H3K9ac antibodies. Waterfall plot of the $\log_2$ (fold-change) in signal intensity in open chromatin (ATAC-seq) peaks and H3K27ac peaks between DMSO and 50 nM CPI-1612 treatment. Blue: peaks with at least 2-fold decrease in signal; red: peaks with at least 2-fold increase in signal. (B) Global effects of CPI-1612 treatment. Top 30,000 peaks for each feature were

ranked based on intensity for DMSO and CPI-1612 conditions. Graphs represent the sum of signal intensity across all peaks. (C) Fraction of peaks located in different genomic regions. Peaks from ATAC-seq or H3K27ac were assigned to the indicated genomic regions using HOMER. All: all peaks identified in DMSO and CPI-1612 conditions; differential: peaks that changed at least 2-fold upon treatment. (D) Genes mapped to differential ATAC-seq or H3K27ac peaks are likely to be downregulated. Genes were assigned to differential ATAC-seq or H3K27ac peaks, and differential expression data (as described in Fig 2) from DESeq2 were plotted. (E) Genes linked to differential features are enriched for ER targets. Genes from (D) were used for GSEA with Hallmark genesets. Top three signatures are shown with NES on the x-axis and adjusted P-value ($P_{adj}$) next to bars. (F) Integrated gene expression and epigenomic features for the *ELF3* locus. Top panel shows annotated genes, and purple boxes show annotated *ELF3* enhancer elements. RNA-seq, H3K27ac ChIP-seq, and ATAC-seq tracks are plotted at the bottom, with H3K27ac and ATAC-seq peaks showing at least a 2-fold decrease marked with a *. The inset shows the differential ATAC-seq peaks in the *ELF3* enhancer.

sites are known to define lineage-specific enhancers in luminal breast cancer cells [29] and several luminal-specific transcription factor motifs (GATA3, TRPS1, etc.) [30] were also significantly enriched in the set of differentially accessibly ATAC-seq sites (Fig 4A). We hypothesized that the enhancers most overtly affected by CPI-1612 were enriched in lineage-specific regulatory elements affecting the expression of luminal gene sets. To test this hypothesis, we used two breast cancer defined gene sets: luminal (Lum(M)-ECJ) to define lineage-specific genes and basal (Bas-ECJ) as a foil for non-luminal genes [31]. GSEA of the differentially expressed genes identified from RNA-seq of CPI-1612 treated MCF7 cells revealed significant enrichment of the luminal gene set for down-regulated genes (p = 1E-10), while the basal gene set showed no significant enrichment (Fig 4B, left, and S8 Fig). Performing the same enrichment analysis with genes mapped from the ATAC-seq analysis again demonstrated a significant enrichment (p = 1E-3) for the luminal set, though only a subset of the differentially accessible peaks could be mapped to genes (Fig 4B, right).

To check whether the differentially accessible sites were occupied by FOXA1 in MCF7 cells we compared these sites using published ChIP-seq data [32]. Confirming what the HOMER analysis predicted, a majority of the differentially accessible sites were bound by FOXA1 in MCF7 cells (Fig 4C). As shown in S5 Fig, 54% of the differential peaks overlapped with a FOXA1 binding site and these peaks were enriched for FOXA1 binding compared to all open peaks (p-value < 1.0E-8, Fisher's Exact test). These loci also contained a significant overlap with H3K27ac, H3K4me1, and ER binding sites. Not surprisingly, given that these sites were predominately located in non-promoter enhancers, there was minimal overlap with H3K4me3 signal.

To explore the translatability of these findings to human breast cancer, we next correlated sites with differential accessibility in response to CPI-1612 treatment in MCF7 cells with known regions of open chromatin in breast invasive carcinoma samples from The Cancer Genome Atlas [23]. We found that the differentially accessible sites upon CPI-1612 treatment were more likely to be accessible in non-basal relative to basal tumors, while randomly selected ATAC-seq peaks showed no difference based on tumor lineage (Fig 4D). Taken together, these data show that CPI-1612 specifically targets intergenic enhancers defined by FOXA1 binding to promote the downregulation of luminal-specific genes in ER positive breast cancer.

## Discussion

CREBBP/EP300 regulate the growth and signaling of normal and cancerous cells by integrating upstream stimuli to tune transcriptional output. There is substantial evidence and compelling rationale to support the key role of these HAT proteins in the treatment of hormone-dependent breast cancer. In this study we show that CREBBP/EP300 HAT inhibition with the potent and selective inhibitor CPI-1612 can inhibit the growth of breast cancer cell lines *in vitro* and impede tumorigenesis *in vivo* at well tolerated doses with demonstrated target engagement. We further demonstrate that CREBBP/EP300 HAT inhibition acts in an

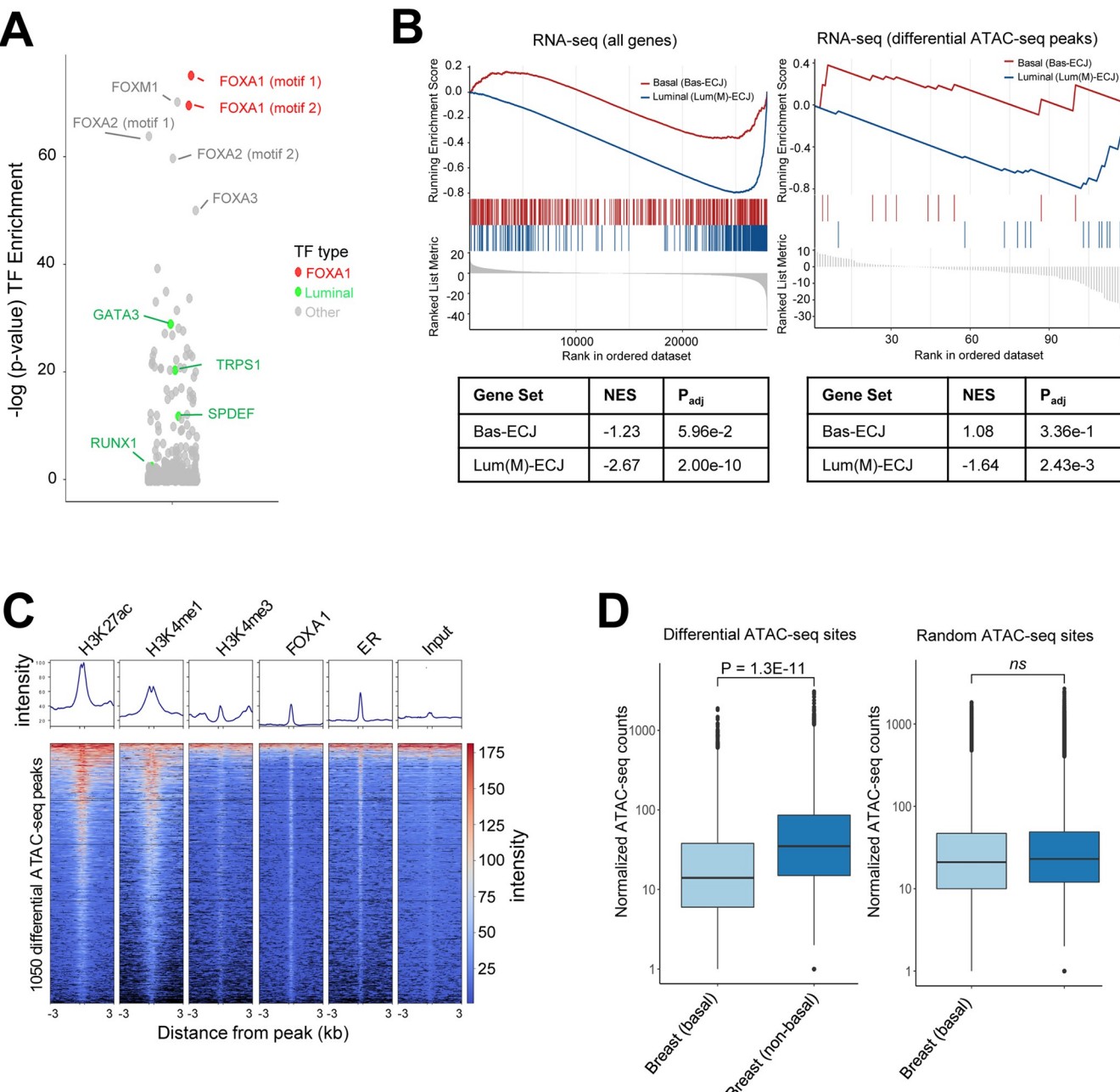

**Fig 4. CPI-1612 targets FOXA1 binding sites that control luminal-specific gene sets in MCF7 cells and breast tumors.** (A) Differential ATAC-seq peaks are enriched for FOXA1 motifs. HOMER motif analysis was used to identify enrichment of transcription factor (TF) motifs in the ATAC-seq peaks that were downregulated at least 2-fold after CPI-1612 treatment, relative to the fraction of all ATAC-seq peaks with binding sites. (B) Downregulated genes and genes mapped to sites of reduced ATAC-seq signal are luminal-specific. GSEA was carried out on all genes (left) or genes mapped to ATAC-seq peaks that changed at least 2-fold (right) using either the Bas-ECJ or Lum(M)-ECJ genesets. NES and $P_{adj}$ values are indicated below the enrichment plots. (C) Differential ATAC-seq peaks are enriched for epigenomic features and TF binding. Published ChIP-seq data for the indicated features in MCF7 cells were plotted for the ATAC-seq peaks that were decreased at least 2-fold after CPI-1612 treatment. (D) Sites of differential ATAC-seq signal are more open in non-basal relative to basal breast tumors. Average ATAC-seq signal across all TCGA samples annotated as either basal or non-basal breast cancer was calculated for each of the differential ATAC-seq peaks described in Fig 3 and was compared to the difference in the average signal across a set of non-differential ATAC-seq peaks. P-values were calculated by unpaired student's t-test with Welch's correction on log-normalized read counts in each peak.

overlapping but nonidentical manner to the standard of care Fulvestrant, supporting its potential development in combination or resistance settings in ER+ breast cancer.

Despite strong evidence linking CREBBP/EP300 to pathological transcriptional programs in cancer, prior to the late 2010s, inhibitors targeting CREBBP/EP300 acetyltransferase activity were limited to cell impermeable substrate analogs (e.g. Lys-CoA), nonselective natural products (e.g. curcumin), or reactive synthetic compounds (e.g. C646) [6]. Work published by Abb-Vie was the first to conclusively demonstrate the feasibility of CREBBP/EP300 acetyltransferase inhibition with drug-like small molecules, of which A-485 was the first example [15]. CPI-1612 was identified as a chemically differentiated acetyl-CoA competitive inhibitor with selectivity over other HAT families, superior potency relative to A-485 in both catalytic domain and full-length biochemical assays, and low nanomolar potency in cell-based target engagement assays [16]. ADME properties are also improved relative to A-485, and it should be noted that CPI-1612 is highly brain penetrant and thus suitable for in vivo studies of CNS malignancies.

Phenotypic profiling of CPI-1612 in breast cancer cell lines showed activity across both ER + and ER- cell lines, with particular sensitivity in several ER+ cell lines. Activity of CREBBP/EP300 HAT inhibition in TNBC cell lines has been noted previously [15], but the biomarkers that predict activity have not been elucidated. However, dependence on the acetyltransferase activity of CREBBP/EP300 in ER+ breast cancer is consistent with the known physical and functional links between CREBBP/EP300 and ER and has been corroborated in another recently published study [33]. However, while Waddell et al. established a link between H3K27 acetylation at enhancers and ER target gene expression, key points were not addressed: comparison of CREBBP/EP300 HAT inhibition with direct ER targeting, the involvement of chromatin accessibility dynamics in transcriptional regulation by CREBBP/EP300, and the features that define the recruitment of CREBBP/EP300 activity to specific loci.

Selective ER degradation by Fulvestrant represents standard of care therapy for ER+ breast cancer. Combination of CPI-1612 with Fulvestrant demonstrated an additive effect on antitumor efficacy of a breast cancer xenograft model, arguing for non-overlapping pharmacology of the two mechanisms. In transcriptional profiling experiments, we noted that while both CPI-1612 and Fulvestrant target the ER transcriptional network, their transcriptional effects at the gene level are distinct. At the dose and timepoint explored, direct ER targeting elicited a modest transcriptional response with affected genes defined by the presence of an ERE. While it is not clear from these experiments whether CREBBP/EP300 HAT inhibition amplifies or accelerates the Fulvestrant transcriptional effect or targets a unique set of genes (or a combination of both), we reasoned that the distinct immediate transcriptional impact may be the result of epigenomic remodeling.

A key observation from our epigenomics experiments is the difference between the broad changes in H3K27ac response and the comparatively more subtle changes observed in chromatin accessibility upon CPI-1612 treatment. Similar to published findings, we observed a global reduction in H3K27 acetylation, with downregulated peaks enriched in enhancers relative to transcriptional start sites [8, 33, 34]. However, significant changes in chromatin accessibility were much less abundant, but were notably enriched with ER target genes. Other studies of CREBBP/EP300 inhibition show similarly modest changes in chromatin accessibility in both multiple myeloma [34] and embryonic stem cells [8], but this phenomenon has not been observed in hormone-dependent cancers. One possible explanation for the limited change in chromatin accessibility is the timepoint used in the experiment–if changes in chromatin accessibility are the result of changes in histone acetylation, one might expect additional loci to be impacted at later timepoints. However, the profound transcriptional effects observed at the early timepoint used in these studies would suggest that many loci are not regulated by changes

in chromatin accessibility upon CREBBP/EP300 HAT inhibition, which is consistent with the work cited above. It is not clear from our data whether the small number of differentially accessible loci are a direct result of acetylation changes, or if they are a consequence of acute transcriptional silencing of the enhancer-gene pair and thus an indirect result of acetylation changes. A recent study in mouse pluripotent stem cells argues that histone acetylation drives the transcription of lineage-specific genes with no changes in global chromatin architecture [35]. However, this does not exclude the possibility that transcription at specific loci is driven by changes in chromatin accessibility.

We reason that the loci showing the most pronounced changes in chromatin accessibility upon CPI-1612 treatment (whether directly or indirectly resulting from loss of histone acetylation) represent the sites that are most dependent on CREBBP/EP300 activity. Consistent with the functional connection of these sites to CREBBP/EP300 activity, differential ATAC-seq peaks were much more likely to be bound by EP300 than differential H3K27ac sites (panel E of S4 Fig). Our observation that these loci tend to be at distal enhancers bound by the pioneer transcription factor FOXA1 provides some insight as to how FOXA1 activates a distinct subset of lineage-defining enhancers [29, 36, 37]. However, while FOXA1 is broadly required for the establishment of open chromatin regions [27, 38], our observation that only a small number of sites show acute loss of chromatin accessibility upon CREBBP/EP300 HAT inhibition may imply the existence of additional layers of chromatin regulation to establish or maintain open chromatin such as histone methylation [39]. Further, it has been shown that ER and FOXA1 show approximately a 50% overlap, arguing for ER binding events that do not directly depend on FOXA1 or CREBBP/EP300 activity and may instead be recruited by features such as "strong" ERE sequences [27, 36]. This differential requirement for FOXA1 and chromatin acetylation/accessibility dynamics for ER chromatin recruitment may contribute to the distinct transcriptional effects of CREBBP/EP300 HAT inhibition relative to direct ER targeting.

From our data it is not known whether FOXA1 directly recruits CREBBP/EP300 activity or whether activity is recruited by additional transcription factors, such as GATA3 or the Mega-Trans complex [40–42]. Notably, members of the core transcriptional regulatory circuitry in MCF7 such as *GATA3* and *ELF3* [24] are represented in both the set of differentially accessible chromatin sites and the set of downregulated genes, arguing that CREBBP/EP300 HAT inhibition impacts these interconnected networks on multiple fronts. The role of acetylation dynamics in FOXA1-mediated enhancer activation may be analogous to the recruitment of HDAC activity to leukemia-specific loci by the pioneer transcription factor PU.1 [43]. Further studies are required to refine the relationship between FOXA1 and CREBBP/EP300 and resolve the temporal relationship among acetylation, chromatin accessibility, and transcription.

This work provides a mechanistic rationale for the exploration of CREBBP/EP300 acetyltransferase inhibition as a therapeutic strategy in ER+ breast cancer. Further studies using the potent and selective inhibitor CPI-1612 will inform clinical development plans and maximize the potential of this approach to address the known limitations of existing therapies that target ER transcription in breast cancer.

## Materials and methods

### Cell lines and compounds

MCF7, T47D, and ZR751 were obtained from ATCC, cultured per supplier's instructions, and used at early passages for experiments. Cells were routinely screened for mycoplasma contamination. Synthesis of CPI-1612 has been described previously [16]. Tamoxifen was obtained from Sigma. Fulvestrant (ICI-182780) was obtained from Sellekchem for in vitro studies.

Fulvestrant (FASLODEX injection/AstraZeneca) was obtained as a dosing solution of 0.25g/5 mL for in vivo studies.

## PRISM cell panel profiling

Pooled screening of barcoded cell lines with a dose titration of CPI-1612 (5 μM, 1.67 μM, 0.6 μM, 0.19 μM, 0.062 μM, 0.021 μM, 0.007 μM, and 0.002 μM) for 5 days was carried out by the Broad Institute PRISM lab according to published protocols [17]. Growth inhibition was assessed using cell barcodes as a proxy for cell number and was expressed as relative growth rate by comparing the cell abundance at the start of the experiment and normalizing to growth rate of the DMSO control. $GI_{50}$ values were calculated using GraphPad PRISM curve fitting to interpolate the concentration at which relative growth rate was 0.5.

## In vitro cell growth assays

For treatments in full serum, cells were plated in 96-well plates and treated in duplicate wells with a dose titration of CPI-1612 for 4 days. For treatments in charcoal-stripped serum, cells were plated in 96-well plates, washed after overnight incubation, and moved to phenol-red free media (Gibco) + 10% charcoal-stripped serum (Gibco) for 2 days. 17-β-estradiol at 100 nM was added along with a dose titration of CPI-1612 for 4 days. Viability was assessed using Cell Titer Glo (Promega), and GraphPad Prism curve fitting was used to fit the data.

## In vivo efficacy studies

All animal studies were carried out at Wuxi AppTec (Shanghai) Co. Inc., with the approval of the Institutional Animal Care and Use Committee (IACUC) of WuXi AppTec following the guidance of the Association for Assessment and Accreditation of Laboratory Animal Care (AAALAC). At the time of routine monitoring, the animals were checked daily for any effects of tumor growth and treatments on normal behavior such as mobility, food and water consumption, body weight gain/loss, eye/hair matting and any other abnormal effect as stated in the protocol. Death and observed clinical signs were recorded on the basis of the numbers of animals within each subset. Animals that were observed to be in a continuing deteriorating condition or their tumor size exceeding 3000 $mm^3$ were euthanized prior to death or before reaching a comatose state.

CPI-1612 was formulated in DMSO/PEG400/$H_2O$, v/v/v, 1/3/6. Stock solutions were prepared for dosing at 0.25 and 0.5 mg/kg at a dosing volume of 10 μL/g twice per day by oral gavage. Fulvestrant was delivered at 1 mg/kg at a dosing volume of 20 μL per mouse once per week by subcutaneous injection.

For efficacy studies, female Balb/c mice at 6–8 weeks old were implanted in the left flank with 17β-Estradiol (0.18 mg) pellets (Innovative Research of America Cat. No.: SE-121, pellet size: 3.0 mm). After 4 days, mice were inoculated subcutaneously in the right flank with 1E7 exponentially growing MCF7 cells in 0.2 mL of PBS/Matrigel at a 1:1 ratio. After average tumor volume reached 150 $mm^3$, mice were randomized for dosing initiation, with 8 mice per group. Animals were monitored for body weight change or other abnormal effects as stated in the approved protocols, and any animal with deteriorating condition or tumor size greater than 3000 $mm^3$ was euthanized. Remaining animals were dosed for 21 days, with tumors measured three times per week in two dimensions using a caliper. Tumor volumes were calculated as V = 0.5a x $b^2$, where a is the short diameter and b is the long diameter.

## Plasma pharmacokinetics analysis

At study endpoint, approximately 50 μL blood was collected from 4 mice in each group and placed in EDTA-2K tubes (1.5 ml tube containing 3 μL of 0.5M EDTA-2K). Anticoagulant blood was centrifuged at 2,000 x g at 4˚C for 15 min. Plasma was stored at -80˚C before analysis. Plasma samples were analyzed by LC/MS/MS, and concentrations of CPI-1612 and Fulvestrant were determined by comparing to standards.

## PBMC preparation and FACS pharmacodynamics assay

At study endpoint, 300–400 μL of whole blood was collected for PBMC isolation from all mice in each group by Ficoll-Paque media density centrifugation. For FACS staining, 2E5 cells were washed twice with DPBS, centrifuged at 400xg for 5 minutes at room temperature (RT), and the pellet was resuspended by flicking. Live/dead viability dye (Biolegend 423114) was diluted 1:1000 in DPBS, and 100 μL was added to the cell pellet. The plate was incubated for 20 min at RT in the dark. Cells were washed twice with 200 μL FACS staining buffer (Ebiosiences 00-4222-26), centrifuged at 400xg for 5 min at RT, and resuspend in 45 μL staining buffer by flicking. Fc Block (BD Biosciences 553142) was added at 5 μL/well and plate was incubated for 5 min at RT in the dark. Staining buffer (50 μL) was added to each well and the plate was incubated for 30 min at 4˚C in the dark. Cells were washed twice with 200 μL staining buffer.

FoxP3 and Transcription Factor Staining Buffer Set (Ebioscience 00-5523-00) was used for intracellular staining. Fixation/permeabilization buffer was diluted to 1x using assay diluent, and 100 μL was added to each well with mixing by pipetting. Plate was incubated for 45 min at 4˚C in the dark, 100 μL of 1x permeabilization buffer (1:10 in water) was added, and plate was centrifuged at 450xg for 5 min. Cells were washed once with 200 μL permeabilization buffer, resuspended in 100 μL staining buffer, and stored at 4˚C overnight. Cells were blocked by adding 100 μL of 1:1000 rabbit gamma globulin (Jackson Immunoresearch) and incubated for 5 min at RT. Cells were washed with 200 μL permeabilization buffer. PE-conjugated rabbit anti-H3K27ac (Cell Signaling 11562S) was diluted 1:20 in permeabilization buffer, and 100 μL was added to cells and mixed by pipetting. Following a 45 min incubation at 4˚C, cells were washed twice with 200 μL permeabilization buffer followed by centrifugation at 450xg for 5 min. Pellet was resuspended in staining buffer and used for FACS analysis. At least 1E4 events (gated on singlet live cells) were acquired for each sample.

## Tumor pharmacodynamics analysis

Tumor samples at study endpoint from same animals as were used for PK analysis were snap frozen in liquid nitrogen and stored at -80˚C. Frozen tumor samples were pulverized using a Covaris tissue pulverizer, returned to liquid nitrogen, and stored on dry ice. For analysis of gene expression, samples were resuspended in 1 mL Trizol (Invitrogen) and homogenized with an Omni Tissue Master homogenizer. Homogenized tumors were centrifuged for 10 min at 4˚C, and supernatant was transferred to fresh tubes. Total RNA was extracted and precipitated according to Trizol manufacturer's instructions. First strand cDNA was prepared using SuperScript IV reverse transcriptase (Invitrogen) according to manufacturer's instructions. Expression of *MYC* in tumors was measured by q-RTPCR with UPL chemistry (Roche, probe #34; F primer 5'tgaattagaatctcgggagtgc3', R primer 5' gagtgagaccccatctcagaa3'), and was normalized to the expression of *ACTB* measured by Taqman chemistry (Applied Biosystems #Hs99999903_m1). RT-PCR data were acquired with a LightCycler 480 (Roche).

For analysis of H3K18ac, pulverized tumors were ground into powder, and 30–40 mg of powder was transferred to a fresh tube. RIPA buffer (Cell Signaling Technology RIPA Buffer (10X) #9806) at 1x (supplemented with 1x complete protease inhibitor cocktail (Roche

#11873580001), 1 mM PMSF (Sigma), and 1:5000 Benzonase (EMD Chemicals #1.01695)) was added at 600 μL per tube. Tumors were homogenized using an Omni Tissue Master homogenizer and incubated on ice for 30 min. NaCl was added to 1M final concentration, and the samples were mixed 3–5 times by pipetting. Samples were sonicated for 2–4 seconds at 40–50% power with a mini probe of a Branson sonifier at setting 3, incubated on ice for 30 min, and centrifuged for 10 min at 12,000xg at 4˚C. Protein concentrations were measured by Bradford assay (Pierce) and samples were diluted in salt/detergent free buffer (20 mM Tris pH 7.5, 1 mM EDTA, 1 mM EGTA, 1x protease inhibitor cocktail, 1mM PMSF) to a salt concentration of 100 mM.

MSD plates (Meso Scale Diagnostics #L15XA-3) were coated with 30 μL per well of capture antibody anti-histone (Millipore #MAB3422) at 4 μg/mL in PBS and incubated overnight. Plates were blocked with 150 μL per well of Blocker A (MSD #R93AA-2) in 1x TBST (TBS + 0.02% Tween-20) with shaking at RT. After Blocker A was removed by flicking and the plate was blotted and washed 1x with TBST, a total of 7.5 μg of lysate was added to each well of a 96-well MSD plate in 100 μL volume of salt/detergent free buffer. Plates were sealed and incubated with shaking for 2 hr at RT. Lysate was discarded at plates were washed 1x with TBST with a plate washer and dried by blotting. Detection antibody (0.125 μg/mL in Blocker A) was added at 25 μL per well (anti-Histone H3 (Cell Signaling #4499) or anti-Histone H3K18ac (Cell Signaling #9675)) and plates were incubated for 60 min at RT with shaking. Sulfo-tag rabbit antibody (MSD #R32AB-1) was diluted to 0.5 μg/mL in Blocker A, and 25 μL was added per well followed by a 1 hr incubation at RT with shaking. Antibodies were discarded by flicking and plates were washed once with TBST using a plate washer. After plates were dried by blotting, 150 μL of Read Buffer (MSD #R92TD-3) was added per well and the plates were read with an MSD SECTOR Imager 2400 according to manufacturer's instructions. For quantification, H3K18ac signal was normalized to total H3 signal.

## Western blotting

Cells were lysed in RIPA buffer (Cell Signaling Technology RIPA Buffer (10X) #9806) at 1x (supplemented with 1x complete protease inhibitor cocktail (Roche #11873580001), 1 mM PMSF (Sigma). Protein concentrations were assessed by Bradford assay (Pierce), and 40 μg of lysate were used for SDS PAGE and Western analysis with visualization on an Odyssey imager (Licor). Primary antibodies were MYC (Cell Signaling #5605), vinculin (Sigma #V9264), and ER-alpha (Millipore #06–935), and secondary antibodies were DyeLight conjugated anti-mouse or anti-rabbit (Licor).

## Bulk RNA-seq

Cells were treated with CPI-1612 for six hours at 5 or 50 nM with or without 100nM Fulvestrant. RNA was extracted with the RNeasy kit (Qiagen), in accordance with the manufacturer's protocol. Samples were treated with DNase and polyadenylated (polyA+) RNA was isolated. Sequencing libraries were constructed using the Illumina TruSeq RNA Sample Preparation Kit (v2). The resulting libraries were sequenced on an Illumina HiSeq 4000, with 9 samples multiplexed per lane. 2x150 base pair paired-end reads using the Illumina TruSeq strand specific protocol, for an expected 20 million reads per sample.

## RNA-seq bioinformatics analysis

Isoform expression of Ensembl transcripts (GRCh38 release 99) was calculated with Salmon version 0.11.3 [44]. Gene-level counts were then imported into R using tximport [45]. Differential expression analysis was performed with DESeq2 [46] to analyze for differences between

conditions, and the set of differentially expressed genes were filtered only for the genes with a | fold-change| greater than 1.5 and a Benjamini-Hochberg FDR adjusted P values less than or equal to 0.05. Venn diagrams were created with BioVenn [47].

## ATAC-seq library preparation and sequencing

MCF7 cells were treated with either 50 nM CPI-1612 or DMSO for 6 hours to profile accessible chromatin. Approximately 10,000 cells for each replicate were profiled using the Omni-ATAC-seq protocol as described [48]. Cells were thawed quickly in a 37˚C rocking bath and 900 μL of ice-cold PBS supplemented with Roche Complete Mini Protease inhibitor was added immediately. Cells were split into two 1.5-ml Eppendorf DNA lo-bind tubes to serve as technical replicates. Cells were centrifuged at 500xg for 5 min at 4˚C, washed once in PBS with protease inhibitor, centrifuged at 500xg for 5 min at 4˚C and supernatant was removed completely using two separate pipetting steps with extreme caution taken to avoid resuspension. The transposition reaction consisted of 20-μL total volume of the following mixture (10 μL 2× TD Buffer, 1 or 0.5 μL TDEnzyme, 0.1 μL of 2% digitonin, 0.2 μL of 10% Tween 20, 0.2 μL of 10% NP40, 6.6 μL of 1× PBS and 2.3 μL of nuclease-free water). Tagmented DNA was purified by QIAGEN MinElute Clean up Kit, PCR amplified and libraries were purified with 1.2X volume of AMPure XP beads. DNA bound to the beads was washed twice in 80% ethanol and eluted in 20 μL of water. Indexed fragments were checked in concentration by qPCR, profiled by Bioanalyzer, equimolarly pooled and sequenced on an Illumina Hi-Seq 4000 with 2 samples multiplexed per lane using 2x150 bp sequencing to a target depth of 60 million reads per sample.

## Bulk ATAC-seq bioinformatics analysis

Paired-end ATAC reads were trimmed to remove Nextera adaptors using Atropos v1.1.28 [49]. Trimmed reads were mapped to the human reference hg38 using BWA mem v0.7.17 [50] with default settings. Aligned bam files were filtered with bamtools [51] to remove a) reads that are not within 2kb on the same chromosome and b) reads with more than four mismatches to reference. Peaks of open chromatin were identified using Genrich [52] in ATAC-seq mode (-j) to adjust for Tn5 shift, using its internal PCR duplicate remover (-r), ignoring mitochondrial (-e chrM) and black-listed regions (-E). The bed file of significant peaks was then processed by HINT [53] to identify footprints with Tn5 bias correction (—atac-seq) and HOMER's [22] findMotifs.pl script was used to identify enriched motifs in the footprints. using Bins Per Million (bpm) with deepTools [54].

## Single cell ATAC-seq

Following 6 hour treatment with DMSO or 50 nM CPI-1612, single MCF7 cells were prepared in accordance with the 10X Genomics manufacturer's protocol. scATAC-seq libraries were constructed using the 10X Genomics Chromium Single Cell ATAC-seq library kit. The resulting libraries were sequenced on an Illumina HiSeq 4000, with 2 samples multiplexed per lane. 2x250 base pair paired-end reads using the Illumina TruSeq strand specific protocol, for an expected 150 million reads per sample (targeting 50,000 reads/cell).

## Single cell ATAC-seq bioinformatics analysis

All raw base call (BCL) files generated by Illumina sequencers were converted to FASTQ files using cellranger-atac mkfastq function. Read filtering and alignment, barcode counting, and identification of transposase cut sites were detected using cellranger-atac count function (cellranger-atac_version 1.1.0) using the CellRanger reference package "refdata-cellranger-atac-

GRCh38-1.1.0" with "gencode.v28.basic" annotations and Signac v(1.3) [55] while filtering samples for cells with at least 3,000 and less than 20,000 fragments per cell. DMSO and CPI-1612 samples were integrated with Harmony v1.0 [56] and transcription factor activity was calculated with chromVAR v1.12 [28] using the JASPAR2020 motif dataset.

## ChIP-seq

Chromatin was prepared by Diagenode ChIP-seq Profiling service (Diagenode Cat# G02010000) using the iDeal ChIP-seq kit for Histones (Diagenode Cat# C01010059).

Chromatin was sheared using Bioruptor® Pico sonication device (Diagenode Cat# B01060001) combined with the Bioruptor® Water cooler for 6 cycles using a 30" [ON] 30" [OFF] settings. Shearing was performed in 1.5 ml Bioruptor® Pico Microtubes with Caps (Diagenode Cat# C30010016) with the following cell number: 1 million in 100μL. An aliquot of this chromatin was used to assess the size of the DNA fragments obtained by High Sensitivity NGS Fragment Analysis Kit (DNF-474) on a Fragment Analyzer™ (Agilent).

ChIP was performed using IP-Star® Compact Automated System (Diagenode Cat# B03000002) following the protocol of the aforementioned kit. Chromatin corresponding to 6μg was immunoprecipitated using the following antibodies and amounts: H3K9ac (C15410004 1μg), H3K27ac (C15410196, 1μg). Chromatin corresponding to 1% was set apart as Input. qPCR analyses were made to check ChIP efficiency using KAPA SYBR® FAST (Sigma-Aldrich) on LightCycler® 96 System (Roche) and results were expressed as % recovery = $2^{\wedge}(Ct\_input-Ct\_sample)$. Primers used were the following: EIF4a2 prom, GAPDH prom, MyoEx2.

The library preparation has been conducted by Diagenode ChIP-seq/ChIP-qPCR Profiling service (Diagenode Cat# G02010000). Libraries were prepared using IP-Star® Compact Automated System (Diagenode Cat# B03000002) from input and ChIP'd DNA using MicroPlex Library Preparation Kit v2 (12 indices) (Diagenode Cat# C05010013). Optimal library amplification was assessed by qPCR using KAPA SYBR® FAST (Sigma-Aldrich) on LightCycler® 96 System (Roche) and by using High Sensitivity NGS Fragment Analysis Kit (DNF-474) on a Fragment Analyzer™ (Agilent). Libraries were then purified using Agencourt® AMPure® XP (Beckman Coulter) and quantified using Qubit™ dsDNA HS Assay Kit (Thermo Fisher Scientific, Q32854). Finally their fragment size was analyzed by High Sensitivity NGS Fragment Analysis Kit (DNF-474) on a Fragment Analyzer™ (Agilent). 2x150 bp paired-end reads were sequenced on an Illumina HiSeq 2500 with a target depth of 60 million reads per sample.

## ChIP-seq bioinformatics analysis

Paired-end ChIP reads were trimmed to remove Illumina adaptors using Atropos v1.1.28 [49]. Trimmed reads were mapped to the human reference hg38 using BWA mem v0.7.17 [50] with default settings. Aligned bam files were filtered with bamtools [51] to remove a) reads that are not within 2kb on the same chromosome and b) reads with more than four mismatches to reference. PCR duplicate reads were removed using Picard (http://broadinstitute.github.io/picard/). Peaks of H3K27ac and H3K9ac were identified using sample-matched input as a control with SICER2 [57] with the gap size (-g) equal to 600 bp. Genomic bigwig tracks were normalized using Bins Per Million (bpm) with deepTools [54]. Proximal genes to annotated ChIP-seq peaks were identified using ChIPseeker [58].

## Supporting information

**S1 Fig. Supplement to Fig 1. A,** Pooled, barcoded cell lines were treated with a dose titration of CPI-1612 for 4 days, and growth inhibition was calculated using depletion of cell line

barcodes relative to initial representation. The concentration at which growth was inhibited to 50% of the untreated cells (GI$_{50}$) was calculated. Red, ER+ cell lines; black: ER- cell lines. **B,** As in **Fig 1A**, except cells were treated in media containing charcoal-stripped serum and added estradiol. Error bars represent SD of 2 replicates. **C,** Change in body weight relative to dosing initiation during treatment with CPI-1612. Error bars represent the SEM at each time point. Data are expressed as H3K18ac signal normalized to total H3 signal in tumor samples from each animal, with mean and SEM shown. P-values were calculated using an unpaired student's t-test. *ns*: p>0.05. **D,** Relative H3K18ac in xenografted MCF7 cells at the endpoint of the study described in **Fig 1B**. **E,** Plasma concentration of CPI-1612 at study endpoint. Data are expressed as mean and SEM across 4 mice, and p-value was calculated using an unpaired student's t-test. **F,** Change in body weight relative to dosing initiation during treatment with CPI-1612 or Fulvestrant. Error bars represent the SEM at each time point. **G,** Plasma concentration of CPI-1612 at study termination for single agent or combination treatment. Data are expressed as mean and SEM across 4 mice, and p-value was calculated using an unpaired Student's t-test. *ns*: p>0.5. **H,** Plasma concentration of Fulvestrant at study termination for single agent or combination treatment. Data are expressed as mean and SEM across 4 mice, and p-value was calculated using an unpaired student's t-test. *ns*: p>0.05.
(TIF)

**S2 Fig. Supplement to Fig 2.** A, Venn diagram of genes down- or upregulated by CPI-1612 or Fulvestrant treatment in MCF7 as in **Fig 2**. Numbers indicate genes down- or upregulated at least 1.5-fold with an adjusted p-value <0.05 in DESeq2 comparisons to DMSO-treated cells. **B,** Summary of GSEA against Hallmark genesets for MCF7, T47D, and ZR751 cells treated with the indicated compounds as described in **Fig 2**. **C,** Enrichment plots for GSEA of RNA-seq data for the HALLMARK_ESTROGEN_RESPONSE_EARLY geneset in MCF7, T47D, or ZR751 cells treated with CPI-1612 low, Fulvestrant, or CPI-1612 low + Fulvestrant as described in **Fig 2**. **D,** Example of differential gene regulation by CPI-1612 and Fulvestrant in T47D and ZR751 cells as described in **Fig 2D**. Values represent the mean and SEM for 3 replicates. P-values were calculated by unpaired student's t-test (*: p<0.05; **:p<0.01;***: p<0.001;****:p<0.0001; ns: not significant). P-values can be found in S6 Data.
(TIF)

**S3 Fig. Supplement to Fig 2.** A, Venn diagrams of genes downregulated at least 1.5-fold in MCF7, T47D, and ZR751 upon treatment with the indicated conditions as in **Fig 2**. **B,** Genomic tracks for the indicated genes showing annotated peaks for ESR1 or EP300 as colored bars. Note that *ELF3* and *HES1* do not have annotated ESR1 peaks. **C,** Fraction of differentially expressed genes from the indicated treatments that were annotated by ChIPseeker as the nearest gene to an EP300 (left panel) or ESR1 (right panel) peak. **D,** Fraction of differentially expressed genes with an annotated Estrogen Response Element (ERE).
(TIF)

**S4 Fig. Supplement to Fig 3.** Comparison of changes in chromatin accessibility and H3K27ac after treatment with CPI-1612. **A,** Venn diagram showing the overlap of all ATAC-seq peaks H3K27ac peaks. **B,** Table summarizing total ATAC-seq and H3K27ac peaks, and peaks with a 2-fold change upon CPI-1612 treatment. **C,** Heatmap of differential ATAC-seq peaks showing that peaks that show the largest change in ATAC-seq signal (k-means cluster #1) are most likely to show a reduction in H3K27ac signal. **D,** Fraction of differential ATAC-seq peaks that are also differential H3K27ac peaks (blue bar), and fraction of differential H3K27ac peaks that are also differential ATAC-seq peaks (orange bar). **E,** Fraction of differential ATAC-seq peaks

and differential H3K27ac peaks that are also occupied by EP300.
(TIF)

**S5 Fig. Supplement to Fig 3.** Western blot of MCF7 cells treated with DMSO, CPI-1612 (0.1 μM), Fulvestrant (1 μM), or Tamoxifen (1 μM) for 24 or 96 hours and probed with the indicated antibodies.
(TIF)

**S6 Fig. Supplement to Fig 4. A,** HOMER motif search of differential H3K27ac peaks upon CPI-1612 treatment. Binding sites for FOXA1 and luminal specific TFs are not enriched in differential H3K27ac peaks relative to all H3K27ac peaks. **B,** Overlap of transcription factor (TF) binding with all identified ATAC-seq peaks or those peaks that showed at least a 2-fold reduction in signal after CPI-1612 treatment. ER and FOXA1 data are as described in **Fig 4C**.
(TIF)

**S7 Fig. Single cell ATAC-seq after CPI-1612 treatment in MCF7 cells. A,** UMAP dimensionality reduction plot of scATAC-seq data colored by treatment for both unintegrated cells and data integrated using Harmony. **B,** Waterfall plot of peaks identified from scATAC-seq data ranked by $\log_2$ (fold-change) for CPI-1612 relative to DMSO. Blue, peaks reduced by at least 2-fold. **C,** UMAP plot as in **A,** colored by predicted FOXA1 activity based on ChromVAR analysis. **D,** Quantification of predicted FOXA1 activity for DMSO and CPI-1612 treated cells. Boxplots depict median and range of FOXA1 (MA0148.4 motif) activity; p-value was calculated with the Mann-Whitney U test.
(TIF)

**S8 Fig. Supplement to Fig 4. A,** Differential expression of genes in the Bas-ECJ or Lum(M)-ECJ gene sets upon treatment with CPI-1612. **B,** Volcano plot of gene expression changes as described in **B**.
(TIF)

**S1 Data. RNA-seq differential expression data for MCF7, T47D, and ZR751 cells treated with CPI-1612, Fulvestrant, or combination.** Source data for S3A Fig.
(XLSX)

**S2 Data. Source data for Fig 1 and S1 Fig.**
(XLSX)

**S3 Data. Source data for Fig 2.**
(XLSX)

**S4 Data. Source data for Fig 3.**
(XLSX)

**S5 Data. Source data for Fig 4.**
(XLSX)

**S6 Data. Source data for S2 Fig.**
(XLSX)

**S7 Data. Source data for S3 Fig.**
(XLSX)

**S8 Data. Source data for S4 Fig.**
(XLSX)

**S9 Data. Source data for S6 Fig.**
(XLSX)

**S10 Data. Source data for S7 Fig.**
(XLSX)

**S11 Data. Source data for S8 Fig.**
(XLSX)

**S1 File. Uncropped westerns from S5 Fig.**
(PDF)

## Acknowledgments

The authors would like to thank colleagues at Constellation for thoughtful discussions and review of the manuscript.

## Author Contributions

**Conceptualization:** Archana Bommi-Reddy, Sungmi Park-Chouinard, Jonathan E. Wilson, Robert J. Sims, III, Andrew R. Conery.

**Data curation:** David N. Mayhew.

**Formal analysis:** Archana Bommi-Reddy, Sungmi Park-Chouinard, David N. Mayhew, Michael J. Steinbaugh, Andrew R. Conery.

**Investigation:** Archana Bommi-Reddy, Sungmi Park-Chouinard, Esteban Terzo, Aparna Hingway.

**Project administration:** Archana Bommi-Reddy, Jonathan E. Wilson, Robert J. Sims, III.

**Software:** David N. Mayhew, Michael J. Steinbaugh.

**Visualization:** Sungmi Park-Chouinard, David N. Mayhew, Andrew R. Conery.

**Writing – original draft:** Archana Bommi-Reddy, Sungmi Park-Chouinard, David N. Mayhew, Andrew R. Conery.

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
