## [Decision Letter · Decision Letter 0]

11 Jan 2022

PONE-D-21-40086CREBBP/EP300 acetyltransferase inhibition disrupts FOXA1-bound enhancers to inhibit the proliferation of ER+ breast cancer cellsPLOS ONE

Dear Dr. Conery,

Thank you for submitting your manuscript to PLOS ONE. After careful consideration, we feel that it has merit but does not fully meet PLOS ONE’s publication criteria as it currently stands. Therefore, we invite you to submit a revised version of the manuscript that addresses the points raised during the review process.

We look forward to receiving your revised manuscript.

Kind regards,

Alessandro Weisz

Academic Editor

PLOS ONE

Journal Requirements:

" All authors are current or former employees and shareholders in Constellation Pharmaceuticals, a Morphosys Company."

We note that you received funding from a commercial source: Constellation Pharmaceuticals, a Morphosys Company

Additional Editor Comments:

The manuscript has been revised by two experts in the field, that both found it of intererest and worth considering for publication, provided the A.s aggree to take into acount the comments and suggestions they raised.

Reviewers' comments:

Reviewer's Responses to Questions

**Comments to the Author**

1. Is the manuscript technically sound, and do the data support the conclusions?

Reviewer #1: Yes

Reviewer #2: Yes

2. Has the statistical analysis been performed appropriately and rigorously? 

Reviewer #1: I Don't Know

Reviewer #2: Yes

3. Have the authors made all data underlying the findings in their manuscript fully available?

Reviewer #1: No

Reviewer #2: Yes

4. Is the manuscript presented in an intelligible fashion and written in standard English?

Reviewer #1: Yes

Reviewer #2: Yes

5. Review Comments to the Author

Reviewer #1: This is a good manuscript and new insight exploring the potential clinical utility of p300/CBP inhibitors is exciting and will be of relevance to the community. The work is high quality and the paper is well written.

- It wasn't clear how many of the differentially regulated H3K27Ac peaks were bound by p300/CBP?

- The authors suggest that certain genes are affected by CPI-1612 but not Fulvestrant (including KLF4, ELF3 and HES1). Are they implying that these genes are bound and regulated by p300/CBP but not ER? This is entirely plausible, but surely the authors can check this with public datasets (i.e. are the genomic regions near KLF4, ELF3 and HES1 bound by p300/CBP but not ER, suggesting a different transcription factor is involved at those genes)?

- The authors claim that less than 5% of differential H3K27Ac peaks show changes in accessibility. However, isn't this simply a timing issue? At 6hr of treatment, I would be surprised if there was sufficient time for chromatin accessibility changes to be observed unless an active closing was engaged. For most sites, depletion of acetylation would result in a passive chromatin closing, which is unlikely to occur by 6hr. If the authors had left the treatment on for longer before conducting ATAC-seq, I suspect that there would be a lot more changes and a substantial fraction of the differential p300/CBP differential regions would have altered accessibility. I don't expect the authors to conduct this experiment, but this possibility (i.e. a short time point doesn't give sufficient time for chromatin closing to occur) should be discussed.

- The data has been deposited but the GEO link is password protected and no password is provided. Since no information is provided in the methods, it's unclear how many replicates of ChIP-seq and ATAC-seq were conducted and since I can't access the data, I am unable to judge whether there is sufficient replicates.

Reviewer #2: In their manuscript, Bommi-Reddy and colleagues demonstrate that the acetyltransferase activity of the ER coactivator CREBBP/EP300 represents a promising therapeutic target in ER+ BC. They propose that, inhibiting the HAT domain of CREBBP/EP300, are able to target the ER transcriptional network. By using a selective inhibitor CPI-1612, they demonstrate that CREBBP/EP300 acetyltransferase inhibition suppresses the growth of breast cancer cell models both in vitro and in vivo by acting in a manner orthogonal to directly targeting ER. CREBBP/EP300 inhibition, according to the authors finding, suppresses ER dependent transcriptional program by targeting lineage-specific FOXA1 bound enhancers.

Despite the study is quite interesting, well-conceived and the generated data of potential interest since they point out the CREBBP/EP300 acetyltransferase activity as a putative target for clinical development in ER+ breast cancer, there are concerns that need to be addressed before the manuscript can be further considered for publication.

- Concerning Supp. Fig. S1A, please include labels related to the different cell line analyzed. Then, there is no indication of the statistical significance in the figures. Please check this point and include in the figure the appropriate statistics. Moreover, I would include an ER negative BC cell line, as control, also in the graph concerning standard growth inhibition assay (Fig. 1A)

- The authors state that CPI-1612 treatment led to dose-dependent reduction in H3K27 acetylation in BC xenograft peripheral blood, demonstrating target engagement at efficacious doses. What about H3K18 acetylation modulation by CPI-1612 treatment?

-Then, considering the RNA-seq results the deepest impact on transcriptome changing in MCF-7 is obtained upon CPI-1612 high+Fulvestrant (Fig. 2A) while the highest impact in GSEA analysis ( NES) is observable following CPI-1612 low+Fulvestrant. Why? The author should comment on that. Moreover, concerning the venn diagram in Supp. Fig.S2A. Are the data referring to MCF-7? I suggest also including a table with all the differentially expressed genes obtained upon transcriptome profiling in the different cell line analyzed as resource for future investigations.and comment on common and specific effect obtained in the manuscript. Then, the authors state that comparison of the enrichment plots for single agent CPI-1612 or Fulvestrant with combination treatment shows enhanced repression upon combination treatment (Fig S2C). This effect is achieved when considering CPI-1612 low or high +Fulvestrant?

- The authors say that combination of ChIP- and ATAC-seq sites identify several key genes, which were down regulated after CPI-1612 treatment, including ESR1. What about ER protein level upon CPI-1612 treatment? Please enhance the quality of panel F of Fig.3. Which were the most statistically significant enriched motif for differential H3K27ac peaks?

-Finally, did the author consider validating their key ATAC-seq finding with additional inhibitors?

Minor points

- I would encourage the authors to enhance the quality of all images along the main text and the supporting information especially for figure labels that result, in some cases, difficult to read.

- There are some typos in the text that need corrections.

6. PLOS authors have the option to publish the peer review history of their article (what does this mean?). If published, this will include your full peer review and any attached files.

Reviewer #1: No

Reviewer #2: No

---

## [Author Response · Author response to Decision Letter 0]

25 Feb 2022

Reviewer #1: This is a good manuscript and new insight exploring the potential clinical utility of p300/CBP inhibitors is exciting and will be of relevance to the community. The work is high quality and the paper is well written.

- It wasn't clear how many of the differentially regulated H3K27Ac peaks were bound by p300/CBP?

Author response

We agree that this is an important calculation that we should add to the manuscript. There are published datasets measuring EP300 binding in MCF7 cells via ChIP-seq. We used the Theodorou et al. data (PMID: 23172872). The proportion of differentially regulated H3K27ac peaks which overlapped with EP300 binding site was 15.8%, while the same calculation for differentially open ATAC-seq peaks was 59.5%. This trend fits with our other observations that the differential ATAC-seq peaks, while fewer in number relative to the differential H3K27ac peaks, are enriched in the more functionally relevant enhancers. We have added these data to the manuscript in S4 Fig and referenced the data in the Discussion section. We thank the reviewer for this suggestion, as we feel it provides further support to our conclusions as to the functional relevance of the sites showing differential chromatin accessibility.

-The authors suggest that certain genes are affected by CPI-1612 but not Fulvestrant (including KLF4, ELF3 and HES1). Are they implying that these genes are bound and regulated by p300/CBP but not ER? This is entirely plausible, but surely the authors can check this with public datasets (i.e. are the genomic regions near KLF4, ELF3 and HES1 bound by p300/CBP but not ER, suggesting a different transcription factor is involved at those genes)?

Author response

Again, we thank the reviewer for this suggestion, as our additional analysis is consistent with the findings reported in the manuscript. Using publicly available datasets, we noted that MYC and GREB1, which are regulated by Fulvestrant or both Fulvestrant and CPI-1612, have annotated peaks for ESR1 and EP300 occupancy, while ELF3 and HES1, which are selectively regulated by CPI-1612, have no annotated ESR1 peaks but do have proximal EP300 peaks. Note that KLF4 did not have annotated peaks for either factor. We have added these data in S3 Fig. Looking more globally, we see that the presence of a proximal EP300 peak does not differentiate genes differentially expressed by Fulvestrant or CPI-1612, but that the presence of a proximal ESR1 peak is more likely for genes differentially regulated by Fulvestrant. We have added these data as panel C of S3 Fig, which nicely complements panel D of S3 Fig showing that genes regulated by CPI-1612 are less likely to have an ERE than those regulated by Fulvestrant. 

- The authors claim that less than 5% of differential H3K27Ac peaks show changes in accessibility. However, isn't this simply a timing issue? At 6hr of treatment, I would be surprised if there was sufficient time for chromatin accessibility changes to be observed unless an active closing was engaged. For most sites, depletion of acetylation would result in a passive chromatin closing, which is unlikely to occur by 6hr. If the authors had left the treatment on for longer before conducting ATAC-seq, I suspect that there would be a lot more changes and a substantial fraction of the differential p300/CBP differential regions would have altered accessibility. I don't expect the authors to conduct this experiment, but this possibility (i.e. a short time point doesn't give sufficient time for chromatin closing to occur) should be discussed.

Author response

This is an excellent point, and one that we did not adequately address in the manuscript. It is definitely possible (or even likely) that we would observe continued closing of chromatin at later timepoints following CREBBP/EP300 HAT inhibition. We focus here on the early timepoint since we already observe acute changes in gene expression (as early as 1 hour in unpublished data), and later changes in chromatin accessibility may be secondary to global transcriptional remodeling rather than direct effects of CREBBP/EP300 HAT inhibition or changes in acetylation. Recent publications are consistent with a global reduction in histone acetylation and a more circumscribed effect on chromatin accessibility following CREBBP/EP300 HAT inhibition, and the authors propose that many genes are regulated by histone acetylation but not chromatin accessibility. It is possible that most genes are regulated by histone acetylation, while a subset is regulated by changes in chromatin accessibility; from our data we cannot differentiate among the possible scenarios. Given the importance of these issues for the understanding of CREBBP/EP300 biology, we have added a further discussion of this point to the Discussion section of the manuscript.

- The data has been deposited but the GEO link is password protected and no password is provided. Since no information is provided in the methods, it’s unclear how many replicates of ChIP-seq and ATAC-seq were conducted and since I can’t access the data, I am unable to judge whether there is sufficient replicates.

Author response

We apologize for the lack of communication regarding access to the embargoed GEO dataset. Please use the following link and password to access the data, which will be released to the public upon publication:

To review GEO accession GSE190163:

Go to https://www.ncbi.nlm.nih.gov/geo/query/acc.cgi?acc=GSE190163

Enter token exotswqchjwnbsl into the box

Regarding the number of replicates, ChIP-seq experiments were carried out with duplicate samples, while ATAC-seq experiments were carried out with single samples. Note that findings from bulk ATAC-seq were confirmed with single cell ATAC-seq carried out independently. 

Reviewer #2: In their manuscript, Bommi-Reddy and colleagues demonstrate that the acetyltransferase activity of the ER coactivator CREBBP/EP300 represents a promising therapeutic target in ER+ BC. They propose that, inhibiting the HAT domain of CREBBP/EP300, are able to target the ER transcriptional network. By using a selective inhibitor CPI-1612, they demonstrate that CREBBP/EP300 acetyltransferase inhibition suppresses the growth of breast cancer cell models both in vitro and in vivo by acting in a manner orthogonal to directly targeting ER. CREBBP/EP300 inhibition, according to the authors finding, suppresses ER dependent transcriptional program by targeting lineage-specific FOXA1 bound enhancers.

Despite the study is quite interesting, well-conceived and the generated data of potential interest since they point out the CREBBP/EP300 acetyltransferase activity as a putative target for clinical development in ER+ breast cancer, there are concerns that need to be addressed before the manuscript can be further considered for publication.

- Concerning Supp. Fig. S1A, please include labels related to the different cell line analyzed. Then, there is no indication of the statistical significance in the figures. Please check this point and include in the figure the appropriate statistics. Moreover, I would include an ER negative BC cell line, as control, also in the graph concerning standard growth inhibition assay (Fig. 1A)

Author response

We thank the reviewer for this comment, which we have addressed with a few modifications to the figures and text that hopefully clarify our conclusions. Rather than cluttering the figure itself, we have only labeled 3 of the ER+ cell lines profiled by standard growth inhibition assays in Figure 1A (the other cell line, T47D, did not behave well in the PRISM assay and was excluded from analysis). Additionally, we have included the source data for the plot as supplementary information (S2_Data), which shows the raw growth rate values and the R2 values of the nonlinear curve fits. Since the data are from a pooled screen of barcoded cell lines, there are not replicate values and thus no statistics can be calculated. We used these data to generate hypotheses about sensitive lineages which could then be followed up with standard growth inhibition assays.

Regarding data for ER negative BC cell lines, we have clarified in the text that we are not arguing that ER+ BC cell lines are more sensitive than ER- BC cell lines. Our only point in showing the whole breast panel is to show that ER+ BC cell lines are phenotypically sensitive, and that not all cell lines are phenotypically sensitive. ER- BC cell lines do seem to be phenotypically sensitive as well, but an investigation into the mechanisms underlying this sensitivity is beyond the scope of this article. We did not include dose response curves for any ER- BC cell line since the focus of the article is about ER+ BC cell lines and we did not want to complicate the narrative.

- The authors state that CPI-1612 treatment led to dose-dependent reduction in H3K27 acetylation in BC xenograft peripheral blood, demonstrating target engagement at efficacious doses. What about H3K18 acetylation modulation by CPI-1612 treatment?

Author response

This is a good suggestion, and in fact we had these data in an earlier draft of the manuscript. We have now included data for H3K18 acetylation modulation in the tumor cells upon CPI-1612 treatment in S1_Fig. The data are noisier than the H3K27ac data, but the trend does show a dose-dependent reduction in H3K18 acetylation.

-Then, considering the RNA-seq results the deepest impact on transcriptome changing in MCF-7 is obtained upon CPI-1612 high+Fulvestrant (Fig. 2A) while the highest impact in GSEA analysis (NES) is observable following CPI-1612 low+Fulvestrant. Why? The author should comment on that. 

Author response

We thank the reviewer for pointing out that difference. Overall treatments of both treatments generate strongly negative enrichment scores and we cannot tell from the statistics whether the difference between the two treatments is significant. If the difference were real, one might imagine that at the higher concentration of CPI-1612, broader transcriptional effects may mask the impact on any single gene set. However, if we view the same GSEA results with the non-normalized enrichment scores or compare p-values for gene set enrichment the differences between the high and low concentrations of CPI-1612 are much more subtle, so we are hesitant to draw any conclusions from the difference. We have added these additional statistics for the GSEA results presented in Fig 2B to Supplementary Information in S3_Data.

Moreover, concerning the venn diagram in Supp. Fig.S2A. Are the data referring to MCF-7? I suggest also including a table with all the differentially expressed genes obtained upon transcriptome profiling in the different cell line analyzed as resource for future investigations.and comment on common and specific effect obtained in the manuscript. Then, the authors state that comparison of the enrichment plots for single agent CPI-1612 or Fulvestrant with combination treatment shows enhanced repression upon combination treatment (Fig S2C). This effect is achieved when considering CPI-1612 low or high +Fulvestrant?

Author response

We apologize for any confusion regarding the figures; we have clarified that the venn diagram is for MCF7 data and have indicated in the supplementary figure legend that the enrichment plots are for CPI-1612 low. We have added a comparison of differential expression across the three cell lines in panel A of S3 Fig, as well as a discussion of the data in the manuscript. Full differential expression data for MCF7, T47D, and ZR751 can be found in S1_Data in the supplementary files, and raw expression data are found on GEO (GSE190163). Data are embargoed until publication but can be accessed by the reviewer. Please go to https://www.ncbi.nlm.nih.gov/geo/query/acc.cgi?acc=GSE190163

and enter token exotswqchjwnbsl into the box.

- The authors say that combination of ChIP- and ATAC-seq sites identify several key genes, which were down regulated after CPI-1612 treatment, including ESR1. What about ER protein level upon CPI-1612 treatment? 

Author response

We did look at the protein levels of both ER-alpha and MYC (two genes that are proximal to differential ATAC-seq peaks that are also downregulated at the mRNA level) and saw that both are reduced upon CPI-1612 treatment. We have added the data as Fig S5, along with a discussion in the text. 

Please enhance the quality of panel F of Fig.3. Which were the most statistically significant enriched motif for differential H3K27ac peaks?

Author response

We apologize for the difficulty in reading the figure. We have increased the font size and resolution of panel F of Fig 3. The top two HOMER motifs identified in the differential H3K27ac peaks were ATF4 and XBP1. For clarity we have added these labels to panel A of S5 Fig. We would note that relative statistical significance of the enrichment of the H3K27ac is much less than the p-values found for FOXA1 in the differential ATAC-seq peaks.

-Finally, did the author consider validating their key ATAC-seq finding with additional inhibitors?

Author response

This is an excellent idea, and one that we would have addressed had the project continued. From published data we expect that CREBBP/EP300 HAT inhibition with distinct inhibitors should be active in ER+ BC and should globally reduce H3K27 acetylation with modest effects on chromatin accessibility. The novel finding of enrichment of FOXA1 sites at differentially accessible loci should be confirmed with additional inhibitors such as A-485 (or optimized compounds from the same series) to exclude the possibility of off-target pharmacology. 

Minor points

- I would encourage the authors to enhance the quality of all images along the main text and the supporting information especially for figure labels that result, in some cases, difficult to read.

- There are some typos in the text that need corrections.

Author response

We have worked to enhance the quality of the images, particularly those with small font sizes. We anticipate that the results should be clearer for readers.

---

## [Editor Report · Decision Letter 1]

2 Mar 2022

CREBBP/EP300 acetyltransferase inhibition disrupts FOXA1-bound enhancers to inhibit the proliferation of ER+ breast cancer cells

PONE-D-21-40086R1

Dear Dr. Conery,

We’re pleased to inform you that your manuscript has been judged scientifically suitable for publication and will be formally accepted for publication once it meets all outstanding technical requirements.

Kind regards,

Alessandro Weisz

Academic Editor

PLOS ONE

Additional Editor Comments (optional):

The revised manuscript is now acceptable for publication
---

## [Editor Report · Acceptance letter]

17 Mar 2022

PONE-D-21-40086R1 

CREBBP/EP300 acetyltransferase inhibition disrupts FOXA1-bound enhancers to inhibit the proliferation of ER+ breast cancer cells 

Dear Dr. Conery:

I'm pleased to inform you that your manuscript has been deemed suitable for publication in PLOS ONE. Congratulations! Your manuscript is now with our production department. 

Kind regards, 

on behalf of

Dr. Alessandro Weisz 

Academic Editor

PLOS ONE